# Early Diagnosis and Treatment of Kidney Injury: A Focus on Urine Protein

**DOI:** 10.3390/ijms252011171

**Published:** 2024-10-17

**Authors:** Duanna Zeng, Bing Wang, Zheng Xiao, Xiongqin Wang, Xiyang Tang, Xinsheng Yao, Ping Wang, Meifang Li, Yi Dai, Xiean Yu

**Affiliations:** 1College of Pharmacy and International Cooperative Laboratory of Traditional Chinese Medicine Modernization and Innovative Drug Development of Chinese Ministry of Education, Jinan University, Guangzhou 510632, China; zengduanna@163.com (D.Z.); xiaozheng1372@163.com (Z.X.); tangxiyangjnu@163.com (X.T.); tyaoxs@jnu.edu.cn (X.Y.); 2NMPA Key Laboratory for Bioequivalence Research of Generic Drug Evaluation, Shenzhen Institute for Drug Control, Shenzhen 518057, China; wangbingszyj@163.com (B.W.); xqwang0219@163.com (X.W.); wangping662@sina.com (P.W.); szlimeifang@126.com (M.L.)

**Keywords:** kidney injury, glomerular filtration barriers, particle size and charge of urine protein, physiological significance

## Abstract

The kidney, an essential excretory organ of the body, performs a series of crucial physiological functions such as waste removal, maintenance of electrolyte and acid–base balance, and endocrine regulation. Due to its rich blood flow and high metabolic activity, the kidney is susceptible to damage. Currently, kidney injury is classified into acute kidney injury (AKI) and chronic kidney disease (CKD), both of which are associated with high rates of morbidity and mortality on a global scale. The current clinical diagnosis of renal injury relies on the assessment of renal filtration function using creatinine and urea nitrogen as “gold-standard” markers. However, the delayed response time, limited specificity, and reduced accuracy of creatinine and urea nitrogen in evaluating kidney injury have significantly hindered advancements in diagnostic methods for kidney injury. Urinary protein is widely utilized as a biomarker for the early diagnosis of kidney injury due to the selectivity of the glomerular filtration system determining whether proteins can pass through the filtration barrier based on their size and charge. Therefore, as a complex biological sample with varying charges and particle sizes, urinary protein is considered an ideal indicator for monitoring the progression of kidney disease. Exploring the relationship between urinary protein and the advancement of kidney injury based on differences in particle size and charge offers a new perspective for assessing and treating such injuries. Hence, we conducted a comprehensive review of 74 relevant studies to gain a thorough understanding of the physiological mechanism and significance of proteinuria production. The aim was to explore the challenges and opportunities in clinical urine protein detection, as well as to discuss strategies targeting glomerular filtration barriers in order to effectively reduce urine protein levels and treat kidney injury, which could provide a new perspective for identifying the progression of kidney injury.

## 1. Introduction

As a crucial excretory organ of the human body [1], the kidney plays vital physiological roles in waste elimination, electrolyte and acid–base balance maintenance, as well as endocrine regulation to ensure internal environment stability. Due to its rich blood supply and high metabolic activity, the kidney is inevitably susceptible to damage. For over 40 years, nephrologists have classified abnormal kidney function into two clinical syndromes: acute kidney injury (AKI) and chronic kidney disease (CKD) [2,3]. In the occurrence and progression of kidney diseases, AKI is closely associated with CKD. AKI serves as a primary factor leading to CKD development, and failure to diagnose and intervene in AKI in time can result in CKD manifestation. During the course of CKD development, AKI often coexists with it, mutually promoting a malignant progression of renal diseases [4]. The grading of both AKI and CKD progression relies on serum creatinine (Scr) levels and glomerular filtration rate (GFR). However, a delayed Scr response time and low accuracy in GFR measurement significantly hinder an accurate identification of the development processes of AKI and CKD. This delay ultimately leads to late diagnosis and treatment for numerous cases of renal diseases that may eventually progress into end-stage renal disease requiring costly kidney replacement therapy or dialysis [5,6,7,8]. Therefore, precise diagnosis of both AKI and CKD has become an urgent global public health issue.

Urinary protein is one of the most widely used biomarkers for the early diagnosis of kidney injury [9]. It serves as the most common, convenient, and cost-effective clinical examination method that reflects kidney function and independently predicts the presence of kidney disease in patients. Considering the structural and physiological characteristics of the glomerular filtration system, the size and charge of proteins determine their ability to pass through the glomerular filtration barrier (GFB). Under normal circumstances, the size restriction and charge selectivity of GFB result in only a limited number of proteins and small molecules being able to pass through freely in urine filtration. However, dysfunction in glomerular filtration can lead to substantial protein leakage from blood into urine [10,11]. The type and quantity of urinary protein dynamically change with the progression of kidney injury. Therefore, exploring the relationship between urinary protein and kidney injury progression based on differences in particle size and charge offers a promising perspective for assessing such injuries [12].

Furthermore, proteinuria plays a direct role in the progression of kidney disease by causing tubular damage through the lumen [13]. When tubular epithelial cells are exposed to plasma proteins, the various chemical attractors, pro-inflammatory cytokines, and extracellular matrix proteins were released from tubular epithelial cells, leading to inflammation and fibrosis in the tubular interstitium. Even in cases where pathological processes primarily affect the glomeruli, proteinuria may still be associated with the development of renal tubulointerstitial disease [14]. Patients who consistently produce large amounts of urinary protein in many types of kidney disease are also often considered to be at high risk for kidney disease progression. A positive association has been observed between a gradual reduction in urinary protein excretion over a relatively short period of time and long-term maintenance of renal function. This finding further emphasizes the importance of minimizing urinary protein excretion as part of a renal protective treatment strategy. Therefore, the strong association between residual proteinuria and renal prognosis suggests that minimizing proteinuria is essential for treating patients with proteinuric CKD.

Consequently, this review aims to examine the physiological mechanism and significance of renal proteinuria while addressing challenges and opportunities associated with urinary protein detection. Additionally, it explores methods targeting the GFB to reduce urinary protein levels for treating kidney injury. Ultimately, these insights aim to provide a novel approach for evaluating and managing progressive kidney injuries, which could provide a new perspective for identifying the progression of kidney injury.

## 2. A New Insight Focused on Protein Particle Size and Charge for Renal Injury Identification from the Perspective of the Physiological Mechanism of the GFB

As previously mentioned, urinary protein is extensively utilized as a biomarker for early kidney injury diagnosis and serves as an independent predictor in diagnosing patients with kidney disease. Furthermore, population-based studies have identified proteinuria as a prognostic indicator for future decline in glomerular filtration rate and the development of end-stage renal disease [15]. So how is renal proteinuria generated and can we gain further insights into its biological aspects of its mechanism?

Under physiological conditions, the GFB as a selective structure based on size and charge [16]. This means that filtration through the barrier is primarily determined by the size and charge of molecules, with larger and negatively charged molecules being selectively prevented from passing through [17]. Specifically, the GFB consists of endothelial cells, basement membrane, and podocyte (as shown in Figure 1). Glomerular endothelial cells (GECs) are intrinsic to the glomerulus and their fenestrae serve as transmembrane pores essential for filtration across the glomerular capillary walls. They play a crucial role in restricting protein passage [18,19]. The glomerular basement membrane (GBM), located between GECs and podocytes, is composed of four extracellular matrix macromolecules: laminins, type IV collagen, nestin, and heparan sulfate proteoglycans. These components intertwine to form a meshwork with pore sizes ranging from 250 to 400 nm. The GBM plays a vital role in selective filtration within the glomeruli [20]. Podocytes are another major component of the GFB, and they form foot processes supported by actin cytoskeleton. Mutations affecting actin or signaling proteins can disrupt this cytoskeletal structure, leading to foot process contraction or disappearance along with proteinuria development [21] (as shown in Figure 2). Structural and functional changes contribute significantly to proteinuria formation [22]. Consequently, due to limitations imposed by size within the barrier, only small amounts of proteins or small molecules smaller than 6 nm can be freely filtered [23], indicating a mechanical barrier with particle size selectivity for filtered substances. Additionally, since there is an abundance of negatively charged glycoproteins on the surface of the GBM, positively charged substances have greater ease in passing through it [24], suggesting a charge-based selectivity for filtered substances.

Due to the varying particle sizes of proteins, the glomerular filtration’s mechanical barrier can only effectively filter a limited number of small-molecular proteins under normal physiological conditions. Additionally, proteins are negatively charged in physiological conditions, and this charge barrier exerts a repulsive effect on them. Consequently, due to the size selectivity and charge characteristics of the glomerular filtration barrier under physiological conditions, only a small amount of small-molecular plasma proteins can be filtered into urine [25]. However, when kidney injury occurs, functional lesions may develop in the glomerular basement membrane, endothelial cells, and podocytes leading to damage in the filtration system. The effectiveness of the charge barrier and particle size barrier is diminished. As a result, there will be alterations in both particle size and charge properties of substances being filtered through the kidneys. This leads to an increase in glomerular filtration protein levels and a decrease in proximal tubule reabsorption protein levels resulting in proteinuria. In other words, the levels of urinary protein vary dynamically with the progression of renal injury.

Therefore, as a complex biological sample with diverse charges and particle sizes present within it, urine is considered an ideal indicator for reflecting kidney disease progression. It has been hypothesized that the proportion of low- and high-molecular-weight proteins in urine would be a better predictor of the type and severity of injury than the amount of protein in urine [26,27]. Currently, clinical urine protein detection primarily focuses on quantifying urinary protein content; however, valuable information regarding kidney injury recognition may remain concealed behind this content alone. Hence, the detailed classification of urine proteins, which integrates differences related to urinary protein particle size and electric charge, will enable us to comprehensively evaluate the progression of kidney injury.

**Figure 1 ijms-25-11171-f001:**
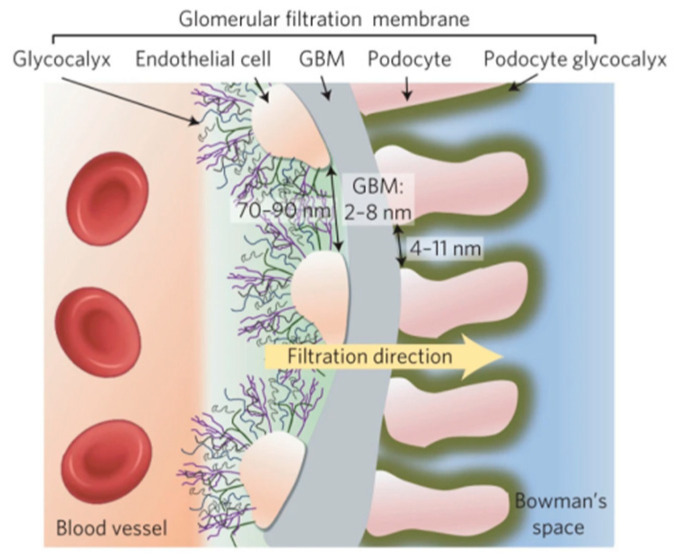
Schematic illustration of glomerular filtration barrier composed of endothelial cells, glomerular basement membrane (GBM), and podocytes [28].

**Figure 2 ijms-25-11171-f002:**
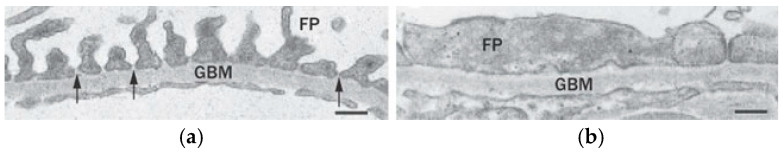
(**a**) A cross-section of a normal glomerular capillary shows the foot processes with interconnecting ultrathin and uniformly wide (40 nm) slit diaphragms (arrows). Scale bar: 200 nm. (**b**) In congenital nephrotic syndrome, the foot processes are lost and the slit diaphragm between the podocytes is absent; the resultant narrow slit lacks a functional filter structure. Scale bar 200 nm. (From the American Society of Nephrology © Lahdenkari, A.T. et al. (2004) [29]. Abbreviations: FP, foot process; GBM, glomerular basement membrane.)

## 3. Challenges and Opportunities in the Detection of Urine Protein

### 3.1. Current State of Clinical Methods for Detecting Urinary Protein

What is the current status of urine protein detection as an important indicator of chronic kidney injury? Current clinical methods for the detection of urinary protein include qualitative, quantitative, and, in some cases, specific protein determination [30].

#### 3.1.1. Qualitative Detection

The qualitative analysis of urine protein involves the determination of urine strip and precipitation methods. Urine test strip detection serves as the fundamental screening test for proteinuria. As the urinary protein concentration increases, the dye indicator (tetrabromophenol blue) undergoes a continuous color change from light green to blue [31]. The binding of tetrabromophenol blue to proteins relies on pH value, but since urine typically has a pH value of 5–6, urine test strips are primarily sensitive to albumin. Furthermore, false positive and false negative results reduce the reliability of qualitative detection [32]. Compared with other detection methods, colorimetric strips offer a low-cost approach that does not require professional operation; however, they often lack accuracy. The precipitation technique measures the turbidity of proteins formed when they are precipitated from solution using sulfosalicylic acid, trichloroacetic acid, or acetic acid and sodium acetate buffers under heating conditions. The turbidity method can detect almost all urinary proteins. However, it is susceptible to interference from various exogenous compounds and therefore rarely used in clinical settings [33].

#### 3.1.2. Quantitative Detection

Quantitative assessment of urinary protein includes the determination of 24 h urinary protein and the protein–creatinine ratio. The gold standard for quantifying urinary protein is the collection and measurement of urine over a 24 h period. Currently, the biuret method is widely utilized to quantify total urinary protein; however, collecting urine over a 24 h period is burdensome, time-consuming, inconvenient for patients, and not entirely foolproof [34]. Measuring the protein–creatinine ratio has gained popularity due to its convenience and time-saving nature in analyzing a single-point urine sample while accounting for changes in urine concentration caused by hydration [35]. There exists a strong correlation between the protein–creatinine ratio and results obtained from 24 h urine collection. Many kidney guidelines now recommend measuring the protein–creatinine ratio in urine collected over a 24 h period [36,37,38]. Currently, biochemical analyzers and other equipment are predominantly employed for clinical quantification of urinary protein; however, this approach proves costly, inconvenient, time-consuming, and necessitates professional operation [39,40].

#### 3.1.3. Specific Protein Detection

The qualitative and quantitative analysis of specific proteins in urine can provide valuable clinical information in certain scenarios. For instance, increased urinary albumin excretion exhibit a strong correlation with the progression of high-risk diseases such as diabetes, hyperglycemia, glomerulonephritis, and cardiovascular disease [41,42]. Moreover, microalbuminuria serves as a clinically significant marker for early diabetic nephropathy and cardiovascular disease. Currently, the identification of albumin and the albumin–creatinine ratio is primarily measured using turbidimetry, radio immunoassays, and enzyme immunoassays; however, some researchers advocate for the widespread adoption of HPLC-based techniques to enhance accuracy [43,44,45,46]. Historically, protein selectivity was determined by assessing immunoglobulin G and transferrin clearance [47], although this is rarely utilized in clinical settings today. Furthermore, urinary immunoglobulin was previously measured through heat coagulation [48]; nevertheless, this technique is no longer employed due to potential false positive results caused by excessive polyclonal light chains [49]. Light chain analysis based on immunoassays now allows for the identification of light urine via urine protein electrophoresis or immunofixation methods [50]. Additionally, there are pregnancy urine tests that employ immunochromatography to detect β-human chorionic gonadotropin levels or determine leukocyte esterase activity.

To conclude, despite being a crucial parameter for early kidney injury assessment, current clinical practice commonly employs various detection methods that possess certain limitations, including prolonged measurement time, intricate procedures, limited sensitivity and detection range, and inadequate sample size, as well as costly instrumentation. Therefore, it is imperative to seek novel breakthroughs.

### 3.2. Utilization of Fluorescence Sensors Based on Particle Size and Charge Offers a Novel Perspective for the Detection of Urinary Protein and the Identification of Kidney Injury Progression

In recent years, there has been rapid development in fluorescence sensing technology, particularly in the field of medical diagnosis. This advancement has provided a new opportunity for detecting kidney injury. Fluorescence sensing technology integrates advanced methods such as molecular recognition and nanotechnology to design nano sensors tailored to individual requirements, thereby achieving accurate detection of kidney injury. One of the unique advantages of fluorescence sensing technology is its ability to distinguish subtle changes in complex samples. This is particularly important as small changes in kidney function during the development of kidney disease are often difficult to detect using traditional methods. Fluorescence sensing technology can capture these subtle changes through accurate measurement of the fluorescence signal. Therefore, as a novel detection technology, fluorescence sensing holds great promise for application in the field of kidney injury detection. Its potential impact on improving diagnostic capabilities and patient outcomes cannot be overstated.

Fluorescence sensors typically consist of fluorophores, linkers, and recognition groups. Fluorophores and recognition groups are linked together by linkers. Once the recognition group identifies a specific analyte, such as a protein or other biological molecules, it will interact with the target analyte in a specific manner. This interaction will then impact the fluorescence intensity, fluorescence lifetime, and other optical characteristics of the fluorophores through various mechanisms. These changes play a crucial role in how the fluorescence sensor responds to the target analyte. In essence, the alteration of optical signals is utilized for detecting the measured object. Subsequently, an array of appropriate statistical methods is employed to analyze these signals for qualitative or quantitative analysis purposes. Fluorescence sensing technology offers several advantages, including high sensitivity, excellent selectivity, rapid response time, ease of operation, and reduced equipment dependency. In the context of extensive research on fluorescence materials and mechanisms, fluorescence sensors have found wide applications in kidney injury studies. They provide new opportunities for early and accurate detection, treatment, and intervention of kidney injuries. For instance, Liu Z.H. et al. [51] synthesized a dual nano indicator that was injected intravenously into a diabetic nephropathy model to measure urine indicators using radio analysis and fluorescence methods. This approach effectively reflected the level of kidney injury. Zhang X.B. et al. [52] successfully developed fluorescent probes for detecting endogenous ^•^O_2_^−^ in living cells and tissues through a dual-model confocal imaging technique. They captured fluctuations in ^•^O_2_^−^ levels during drug-induced nephrotoxicity. Luo Z et al. [53] reported an economical and efficient ratio probe based on albumin-induced aggregation decomposition for rapid on-the-spot detection of albumin using the staging index of chronic kidney disease. Huang J.G. et al. [54] synthesized an activated double-stranded reporter gene (ADR) for real-time in vivo imaging of contrast media-induced AKI (acute kidney injury). Their method detected contrast media-induced AKI at least 8 h earlier than clinical detection using near-infrared fluorescence, etc. This provides novel opportunities for real-time non-invasive monitoring of kidney function at the molecular level.

Furthermore, based on the biological characteristics of traditional indicators of urinary protein production, a detailed classification was conducted, taking into account particle size and charge. Subsequently, by utilizing fluorescent array sensors, microscopic molecular recognition was translated into macroscopic visible optical signals to amplify the differences in traditional indicators. This approach aims to explore additional information associated with these indicators and provide a novel perspective for kidney injury assessment. Building upon this foundation, Yu X et al. [55] focused on the size and charge of urinary proteins and developed a multi-channel biosensor array (polydopamine polyethyleneimine, PDA-PEI) capable of identifying proteins in urine that exhibit varying compositions during different stages of kidney injury progression. By considering both protein particle size and charge difference as factors, they expanded the original three variables caused by protein charge to six variables. The formation of a characteristic fluorescent pattern in urine enabled the instantaneous and accurate identification of kidney injury progression through signal amplification. This strategy not only introduces new perspectives for identifying kidney injury progression based on size and charge but also enhances sensor arrays’ resolution by increasing the number of sensor elements, thereby expanding their potential applications across various diseases. Bai X et al. [12] also investigated the disparities in urine protein particle size and charge during the progression of drug-induced kidney injury. They ingeniously developed a fluorescent array sensor (amino headgroup-terminated gold nanoparticles and fluorophore-modified proteins, AmiNP-FMPs) which possesses remarkable capabilities for detecting intricate biological samples and combined it with multivariate statistical analysis methods to achieve highly sensitive, rapid, and convenient identification of drug-induced kidney injury processes. Similarly, Sun K et al. [56] focused on discerning the discrepancies in urine protein particle size and charge throughout the advancement of drug-induced kidney injury. They devised a novel multi-channel sensor (gold nanoparticles–polyethyleneimine/fluorophore-labeled proteins, AuNPs-PEI/FLPs) to detect the specific fluorescence response mode generated by urine affected by drug-induced kidney injury. By employing multivariate statistical analysis techniques, they accurately identified different types of kidney injuries induced by various drugs.

Therefore, in the presence of complex and diverse urinary proteins, it is not feasible to accurately comprehend the progression of kidney injury solely by measuring urinary protein content. The comprehensive categorization of urine proteins is predicated on their biological characteristics: particle size and charge. The utilization of fluorescence sensors facilitates the conversion of microscopic biological information into macroscopic visible optical signals, thereby amplifying the disparities in traditional indicators. This enables the rapid monitoring of complex urine proteins and facilitates the expeditious and convenient non-invasive identification of the kidney injury process.

## 4. Common Strategies and Mechanisms for Targeting the GFB to Mitigate Urinary Protein Excretion

What are the long-term implications of treating proteinuria? And what are the common clinical methods and mechanisms of targeting the GFB to reduce proteinuria? According to the Kidney Disease: Improving Global Outcomes (KDIGO) 2021 Clinical Practice Guidelines, proteinuria independently promotes progression and serves as a key marker of “sustained kidney injury” [57]. Numerous prospective clinical studies have demonstrated a close relationship between the level of proteinuria and the rate of renal failure progression in both diabetic and non-diabetic chronic kidney disease patients [58,59,60,61,62]. Leakage of plasma proteins into the glomerular filtrate directly damages the tubular epithelium, resulting in loss of tubular function and interstitial scarring [13,63,64]. Therefore, it is important to consider proteinuria not only as an indicator of glomerular injury severity but also as a direct microtubule toxicity factor [65]. The amount of urinary protein correlates with the rate at which renal function declines; thus, treating proteinuria is crucial for delaying kidney disease development and improving prognosis. Most kidney diseases are characterized by dysfunction in the GFB, making it a potential structural target for novel kidney therapies [66]. Furthermore, based on the physiological mechanism described above regarding urinary protein production, abnormality in the GFB leads to its formation. Therefore, we will provide examples of two commonly used clinical drugs and some Chinese medicines that target the GFB to effectively reduce proteinuria and treat kidney injury.

Effective methods for the treatment of renal proteinuria in clinical practice are still lacking. Current treatments primarily focus on addressing the underlying etiology, with commonly used drugs including renin–angiotensin system inhibitors (RASIs) like angiotensin-converting enzyme inhibitors (ACEIs), angiotensin II receptor blockers (ARBs), and immunosuppressants like glucocorticoids; however, these medications may have certain side effects. RASIs are frequently utilized as the initial pharmacological intervention in the management of hypertensive nephropathy, diabetic nephropathy, and other forms of nephropathy. Immunosuppressants also play an important role in the treatment of kidney disease. ACEIs and ARBs target the GFB to improve the selective permeability of the membrane. ACEIs and ARBs can block AngII to reduce urinary protein excretion by improving selective permeability of the GFB [67]. Additionally, renin–angiotensin–aldosterone system inhibitors (RAASi) have been reported to decrease proteinuria in nephrotic animal models while increasing expression of integrin a3 in podocyte basement membranes and reducing podocyte loss [68]. The podocytes of the GFB play a pivotal role in the pathogenesis of proteinuria and nephrotic syndrome [69]. Glucocorticoids are considered a first-line therapy for nephrotic syndrome due to their potent anti-inflammatory and immunosuppressive effects, which are mediated by direct interaction with podocytes. Upon binding to its receptor in the cytoplasm, glucocorticoids translocate into the nucleus where they bind to specific response elements. Notably, human podocytes express glucocorticoid receptors. In vitro studies have demonstrated that co-incubation of dexamethasone with human podocytes leads to increased expression of Nephrin, a crucial slit diaphragm protein in podocytes, while reducing the levels of vascular endothelial growth factor, suggesting direct effects on these cells [70]. Furthermore, dexamethasone has been shown to ameliorate purinomycin-induced injury in cultured podocytes by promoting the repair of damaged cytoskeletal proteins. Additionally, it exerts protective effects against apoptosis by inhibiting p53 expression and downregulating the pro-apoptotic gene Bax while upregulating the anti-apoptotic gene Bcl-xL. In conclusion, glucocorticoids exert anti-inflammatory and immunosuppressive effects and directly safeguard podocyte integrity through the inhibition of apoptosis and preservation of morphological characteristics. These effects demonstrate significant efficacy in the management of renal proteinuria and preservation of kidney function.

As a crucial component of traditional medicine, traditional Chinese medicine has garnered significant attention in the field of biomedicine. It possesses unique advantages and prospects for diagnosing and treating early kidney injury, particularly diabetic nephropathy (DKD). For example, tripterygium wilfordii is an example of a traditional Chinese medicine commonly used for the treatment of DKD, specifically targeting podocytes, which are critical components of the GFB. Tripterygium wilfordii protects podocytes through various means such as anti-inflammation, antioxidation, and anti-apoptosis mechanisms while stabilizing podocyte cytoskeletons; it plays an essential role in reducing proteinuria and improving DKD outcomes. For example, triptolide protects podocytes from apoptosis and reduces proteinuria by enhancing the phosphorylation of NF-κB inhibitory protein α, suppressing NF-κB activation, downregulating the growth arrest and DNA damage-related apoptotic pathway, as well as inducing the expression of protein 45B and caspase 3 [71]. Additionally, triptolide can protect podocytes from PAN-induced cytoskeleton damage and abnormal expression of cressin and podocytin by anti-oxidation, inhibiting inflammation and restoring RhoA activity; it can also effectively reduce proteinuria and improve renal function [72]. In addition, Yu et al. [73] discovered that curcumin effectively suppresses the expression of ATF4, p-eIF2α, and CHOP in podocytes while inhibiting intracellular Ca^2+^ upregulation. This suggests that curcumin intervenes in the endoplasmic reticulum stress pathway to inhibit podocyte apoptosis. Qin et al. [74] found that berberine can enhance fatty acid oxidation, reduce lipid accumulation, improve mitochondrial damage, and inhibit podocyte apoptosis by targeting the peroxisome proliferator-activated receptor γ-coactivator 1α (PGC-1α). BAI et al. demonstrated that Huangkui capsule (HKC) could ameliorate the significant loss of nephrin and podocin expression caused by doxorubicin, indicating its potential to maintain the expression of these two crucial membrane proteins and preserve the structural and functional integrity of podocytes [12]. Numerous studies have focused on traditional Chinese medicines for treating kidney injury by improving the GFB; however, this section only provides a brief overview with selected examples. Nonetheless, current research in this field still faces limitations such as an abundance of basic studies and unclear mechanisms; thus, further investigations are necessary to elucidate the specific mechanisms underlying traditional Chinese medicines’ effects on the GFB and their clinical applications.

In summary, we provide examples of RASIs, such as ACEIs and ARBs, as well as examples of immunosuppressants, like adrenal glucocorticoids, both of which are commonly used clinical drugs. We briefly elucidate their mechanisms in targeting the GFB system to effectively treat proteinuria. These drugs are currently widely used and have been proven through clinical observation and experimental research in modern medicine to significantly reduce proteinuria. However, certain aspects regarding their usage remain unclear, including how to minimize potential side effects on the body. It is imperative that future experimental research and clinical practice focus on finding answers to these questions. Additionally, there are instances where Chinese medicines specifically target the GFB for treating proteinuria. We anticipate further exploration into this promising new target of the GFB, which will lay a solid foundation for developing novel therapies that directly intervene in the GFB and address various kidney disease issues.

## 5. Conclusions and Prospects

Currently, most clinical urine protein detection methods focus merely on the concentration of urine protein, overlooking valuable information that could aid in the identification of kidney injury. The complexity of urinary protein type and quantity means that measuring urinary protein content alone cannot fully and accurately reflect the progression of kidney injury. To comprehensively evaluate kidney injury progression, an integrated analysis of differential characteristics such as particle size and charge of urinary proteins holds potential. In terms of detection, a careful classification of urine protein based on biological characteristics including particle size and charge can magnify the differences in traditional indicators to achieve more accurate early detection of kidney injury. As for treatment, targeting the GFB to reduce proteinuria based on the biological characteristics of urinary protein production represents an innovative and promising approach in nephropathy treatment. Further studies are needed to better understand the complex interaction between the GFB and urinary protein, with the aim to identify more potential therapeutic targets for developing targeted, effective, and safer drugs for treating kidney damage. Therefore, in future studies, whether in the construction of detection methods or the assessment of treatment outcomes, information on particle size and charge can be further deepened on the basis of the total amount of urine protein. This deeper understanding has the potential to enhance accurate diagnosis and improve treatment effectiveness.

## Data Availability

Data are contained within the article.

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
