# Peer review of "Early Diagnosis and Treatment of Kidney Injury: A Focus on Urine Protein"

_ijms, 2024, doi:10.3390/ijms252011171_

Round 1
Reviewer 1 Report (Previous Reviewer 2)
Comments and Suggestions for Authors
Thank you for the possibility to evaluate the manuscript: Early diagnosis and treatment of kidney injury: a focus on the urine protein.
The Authors have addressed almost all my suggestions and doubts.
Still for me it is a mistake that authors mention in details the role of traditional Chinese medicine and do not briefly discuss other commonly used immunosupressants as calcineurin inhibitors, mycophenolate mofetil or cyclophosphamide in proteinuria (glomerulonephritis) treatment. It is not improving the clarity in discussion.
I do not agree that for instance calcineurin inhibitors have only the influence on GFB by suppressing the immune response.
Author Response
Answer: Thank you very much for your second review and the attention you have given to our manuscript. We are pleased to learn that you believe we have addressed most of your suggestions and questions. At the same time, we highly value your concern about the inadequate discussion on the application of other commonly used immunosuppressants in the treatment of proteinuria (glomerulonephritis), particularly the lack of thorough exploration of calcineurin inhibitors, mycophenolate mofetil, or cyclophosphamide in this context. This is primarily because these immunosuppressants do not directly target glomerular filtration barriers (GFB) to reduce proteinuria. Instead, their mechanisms of action primarily involve inhibiting the activation and proliferation of immune cells, thereby reducing the immune-inflammatory response in the kidney and indirectly lowering proteinuria. In the original manuscript, we mainly discussed immunosuppressants that directly target GFB to reduce proteinuria, such as ACE inhibitors (ACEI), angiotensin receptor blockers (ARB), and glucocorticoids. Regarding the aforementioned immunosuppressants, we did not delve into their specific mechanisms for reducing proteinuria because their modes of action are not directly targeted at GFB.
In summary, we are very grateful for your review and valuable comments. We look forward to receiving your further review comments and appreciate your continued support.
Reviewer 2 Report (New Reviewer)
Comments and Suggestions for Authors
The manuscript by Zeng D. et al. is an important summary and a valuable source of knowledge about the need to understand and distinguish the diagnostic role of total protein and individual proteins contained in urine. Despite the well-justified possibilities of using urine as a target clinical material for the kidneys and the diagnostic benefits resulting from the determination of individual proteins of different sizes and charges in urine, contemporary laboratory practice is based mainly on the assessment of the total amount of urine protein. Overcoming the main obstacle, which is undoubtedly the high cost of simultaneous determination of a panel of proteins in urine, may provide objective diagnostic information for making strategic therapeutic decisions in the ongoing progression of changes in glomerular filtration barriers. The new technological possibilities presented in this work may facilitate the solution of this problem.
Author Response
Answer: Thank you very much for your thorough review of our manuscript and your valuable comments. Your feedback not only acknowledges the value of our work but also highlights the limitations in current clinical practice and directions for future research.
Firstly, we fully agree with your perspective that urine holds great potential as a target material for diagnosing kidney diseases, and the measurement of individual proteins in urine, based on their different sizes and charges, can provide abundant diagnostic information for clinical practice. This understanding is the starting point of our research and an important impetus for us to further explore the application of new technologies to optimize urine protein analysis.
Secondly, you mentioned that current laboratory practice primarily relies on the assessment of total urine protein, which indeed reflects the limitations of current clinical practice. We recognize that, given the complexity of urine protein types and quantities, relying solely on the measurement of urine protein content cannot comprehensively and accurately understand the development process of renal injury. Instead, a detailed classification of urine proteins based on the biological characteristics of traditional indicators of urine protein production can amplify the differences among these indicators and provide more biological information for rapidly and conveniently identifying the progression of renal injury.
Furthermore, the high cost of simultaneously measuring multiple proteins in urine, as emphasized by you, is one of the major obstacles hindering its widespread clinical application. We are deeply grateful for your pointing out this issue, as it is indeed a challenge that needs to be focused on and addressed in our research. In our future studies, we will strive to find more economically feasible solutions, aiming to maintain diagnostic accuracy while reducing detection costs.
Lastly, your recognition of the potential of our proposed new technologies is a great encouragement to our work.
In summary, we are very grateful for your review and valuable comments, which not only acknowledge our work but also provide direction for our subsequent research. We will continue to dedicate ourselves to the study of urine proteins, aiming to provide new ideas and solutions to address the issues in identifying the progression of renal injury in current clinical practice.
This manuscript is a resubmission of an earlier submission. The following is a list of the peer review reports and author responses from that submission.
Round 1
Reviewer 1 Report
Comments and Suggestions for Authors
Proteinuria is not a valuable marker for detection of early injury. It is not possible to distinguish if there is a active injury or chronic changes based only on proteinuria.
*Added by editor - additional information from the reviewer:
I agree that serum creatinine is not a best robust marker for kindey injury but urinary protein is not either. There are cases when AKI happens wihtout proteiunuria. Proteinuria might be a merker of glomerular barrier injury but it is not possible if the injury inside kidney is active or chronic. Therefore proteinuria might serve as a risk factor of progression like in IgA nephropathy but is not a sensitive enough to be a marker for early diagnosis.
I have no remarks.
Author Response
Reviewer: 1
Proteinuria is not a valuable marker for detection of early injury. lt is not possible to distinguish if there is a active injury or chronic changes based only on proteinuria.
*Added by editor -additional information from the reviewer.
I agree that serum creatinine is not a best robust marker for kindey injury but urinary protein is not either. There are cases when AKl happens wihtout proteiunuria. Proteinuria might be a marker of glomerular barrier injury but it is not possible if the injury inside kidney is active or chronic. Therefore proteinuria might serve as a risk factor of progression like in lgA nephropathy but is not a sensitive enough to be a marker for early diagnosis.
Answer: We thank the reviewer for the valuable suggestion. First, we acknowledge the limitations of proteinuria and serum creatinine in the early detection of kidney injury as pointed out by the reviewer and editor. In clinical practice, staging of kidney injury relies on complex and potentially invasive methods, such as GFR calculation (involving blood creatinine) and urinary protein detection, which necessitate blood and urine samples from the patient and take multiple physiological parameters into account. However, these traditional markers lack sensitivity and specificity, particularly in distinguishing acute from chronic kidney injury and achieving early diagnosis.
Nevertheless, the significance of urinary protein as a widely used biomarker in the early diagnosis of kidney injury, especially in diabetic nephropathy, cannot be overlooked. Feedback from clinicians also indicates that urinary protein is often a crucial indicator of kidney injury; many diagnoses of kidney diseases begin with an increase in urinary protein levels before calculating filtration rates to determine the severity of kidney injury. We firmly believe that urinary protein holds significant potential in the diagnosis of kidney injury. However, current methods have certain limitations, including lengthy measurement times, complex operations, insufficient sensitivity, and a limited detection range. These limitations not only impact the efficiency of detection but also have the potential to adversely affect the accuracy of diagnostic results.
To address these limitations, our team has innovatively proposed a strategy for the classification of filtered proteins based on particle size and charge. This strategy is informed by the mechanical and charge barrier properties of the glomerular filtration barrier. Additionally, we have developed a novel urine protein fluorescence sensor , leveraging the advantages of fluorescence sensors in detecting complex biological samples. This method transforms microscopic biological information into macroscopic visible optical signals, amplifying the differences in traditional indicators and enabling us to explore more biological information. The rapid response characteristics of optical signals allow us to monitor protein changes in complex urine more quickly and conveniently. Furthermore, this approach provides a powerful tool for identifying kidney injury processes. We firmly believe that an in-depth exploration of the internal relationship between urinary protein and kidney injury will help further tap into the huge potential of urinary protein for accurately diagnosing kidney injury. Our team has been making unremitting efforts in this regard, with all published results described in our article (Lines 255-284). In our latest research results soon to be published, we have combined accumulated findings with artificial intelligence algorithms and successfully applied them to rapidly identify clinical chronic kidney disease urine samples. This not only simplifies the process of staging kidney injuries but also opens up new possibilities for early detection and accurate treatment of kidney injuries.
In conclusion, we extend our sincere gratitude to the reviewers and editors for their careful attention and valuable suggestions on our research. We eagerly anticipate further research and validation to facilitate the widespread application of this new approach in clinical practice, thereby making significant contributions to the prevention and treatment of kidney diseases.
Reviewer 2 Report
Comments and Suggestions for Authors
Thank you for the possibility to evaluate the manuscript: Early diagnosis and treatment of kidney injury: a focus on the urine protein.
The subject the authors would like to discuss is very interesting, but in my opinion, there are speculations, not scientific data.
Currently, kidney injury is classified into AKI and CKD – I do not agree with such comment. For instance - proteinuria, hematuria and accompanying disease syndromes should be also treated as kidney injury.
The sentence - proteinuria plays a direct role in the progression of kidney disease by causing tubular damage through the lumen – is naive and does not provide any new information.
Also this statement is not true - Patients who excrete large amounts of urinary protein are considered a high-risk group for kidney disease progression, even when receiving what is considered optimal treatment. – If patients response to the treatment, it is not dangerous – for instance in most cases of idiopathic nephrotic syndrome in children.
The title of second section should be rewritten “A new insight focused on protein particle size and charge for renal injury identification from the perspective of physiological mechanism of GFB” – in the section were described the data known so far, without any novelty.
The sentence: Traditionally, protein selectivity has been determined by assessing immunoglobulin G and transferrin clearance [46], - in my opinion it was not traditionally. It was proposed more than 55 years ago, but not recommended.
The authors underline, that in recent years, there has been rapid development in fluorescence sensing technology, particularly in the field of medical diagnosis. The utilization of fluorescence sensor facilitates the conversion of microscopic biological information into macroscopic visible optical signals – the examples should be provided.
For me it is a mistake that authors mention traditional Chinese medicine and do not discuss other immunosupressants as calcineurin inhibitors, mycophenolate mofetil or cyclophosphamide in proteinuria (glomerulonephritis) treatment. Rituximab is also not mentioned. Also from the other side for biomedical substances, they only have few literature references.
Authors anticipate further exploration into this promising new target of the GFB, which will lay a solid foundation for developing novel therapies. In fact there are speculations not supported by any evidence in this manuscript.
Author Response
Reviewer: 2
1. The subject the authors would like to discuss is very interesting, but in my opinion, there are speculations, not scientific data.
Answer: We thank the reviewer for the valuable suggestion. We acknowledge your concerns and would like to provide further clarification on our team's research content and data support. Our focus was on addressing the challenging and crucial topic of proteinuria in early kidney injury detection, while recognizing the limitations of traditional markers such as proteinuria and serum creatinine in terms of sensitivity and specificity. However, our study is not merely speculative; it is based on a comprehensive review of the literature, feedback from clinicians, and an understanding of the physiological mechanisms of the glomerular filtration barrier (GFB), leading us to propose innovative solutions. In this paper, we delve into the mechanical and charge-barrier properties of GFB, explaining how these properties impact the formation of proteinuria. Building upon this understanding, along with insights into inherent particle size and charge biology in urine protein production, we have developed novel urine protein fluorescence sensors that amplify differences in traditional indicators by transforming microscopic biological information into macroscopic visible optical signals. This allows for a more thorough exploration of biological information. By leveraging the rapid response characteristics of optical signals, we are able to monitor changes in urinary proteins more quickly and conveniently within complex urine samples. This provides a powerful tool for identifying processes related to kidney injury.
We firmly believe that further exploration of the internal relationship between urinary protein and kidney injury, based on particle size and charge in addition to traditional indicators, will help us unlock the vast potential of urinary protein in accurately diagnosing kidney injury. Our team has been making relentless efforts towards this goal, with our published results in this field detailed in the review (Lines 255-284). In our upcoming research publication, we have successfully integrated artificial intelligence algorithms into the detection of clinical chronic kidney disease urine samples. This not only simplifies the process of kidney injury staging but also opens up new possibilities for early detection and accurate treatment of kidney injury. We provide comprehensive experimental data and analysis to support our findings and conclusions, firmly grounding our research in scientific evidence rather than pure speculation. We eagerly anticipate further research and validation to facilitate the widespread application of this new direction in clinical practice, thereby making significant contributions to preventing and treating kidney diseases. Additionally, we welcome criticism and further verification from peer experts as a means to continuously improve and enhance our research results.
- Currently, kidney injury is classified into AKl and CKD -I do not agree with such comment. For instance - proteinuria, hematuria and accompanying disease syndromes should be also treated as kidney injury.
Answer: In current medical practice, AKI and CKD serve as the primary classification framework for kidney injury. These classifications are based on strict medical guidelines designed to assist physicians in promptly and accurately assessing a patient's kidney function status and tailoring treatment accordingly. The proteinuria and hematuria you mentioned are specific clinical manifestations of renal dysfunction, which play a significant role in the diagnostic process of kidney injury by providing doctors with key diagnostic information.
- 3. The sentence - proteinuria plays a direct role in the progression of kidney disease by causing tubular damage through the lumen -is naive and does not provide any new information.
Answer: I acknowledge the reviewer's concern regarding the amount of information and depth of the sentence. This statement has been referenced in article 13, citing the study by Abbate, Zoja, and Remuzzi (2006). My original intention in referencing this study was to provide a brief overview of the direct role of proteinuria in kidney disease progression, particularly highlighting its key finding on tubular damage through the lumen. I understand that some may find it straightforward and lacking new information. However, I would like to clarify that this sentence does not stand alone in the text; rather, it serves as a pivotal argument in the introduction, aiming to succinctly summarize proteinuria's significant role in kidney disease progression. In subsequent paragraphs, I delve into specific mechanisms by which proteinuria leads to tubulointerstitial inflammation and fibrosis by inducing various chemical attractors, pro-inflammatory cytokines, and extracellular matrix proteins from tubular epithelial cells. Additionally, I discuss the potential association of proteinuria with renal tubulointerstitial disease development even when glomerulopathy is dominant pathologically and highlight a positive correlation between levels of proteinuria and risk of kidney disease progression. Together these contents constitute an exploration into the complex mechanism of proteinuria in nephropathy progression.
Therefore, I believe that the original sentence holds a specific status and role within the text. As a concise expression within the introduction section, it provides necessary groundwork and guidance for further elaboration on subsequent content. Of course, I also recognize that academic research requires constant pursuit of innovation and depth; thus moving forward with my research and writing endeavors will continue striving to uncover new information and perspectives to enrich my research content.
- Also this statement is not true - Patients who excrete large amounts of urinary protein are considered a high-risk group for kidney disease progression, even when receiving what is considered optimal treatment. - lf patients response to the treatment, it is not dangerous – for instance in most cases of idiopathic nephrotic syndrome in children.
Answer: I understand that you may believe that in certain specific conditions (such as idiopathic nephrotic syndrome in children, etc.), patients are not necessarily at high risk if they respond well to treatment despite excreting a large amount of urinary protein. Indeed, the idiopathic nephrotic syndrome in children you mentioned is a good example where many patients can significantly reduce urinary protein excretion and improve kidney function after receiving appropriate treatment. However, my statement is intended to emphasize that generally, in a broader population of patients with kidney disease, consistently excreting large amounts of urinary protein predicts a higher risk of disease progression, even with currently considered optimal treatment. I want to stress the point that "Consistently excreting large amounts of urinary protein generally predicts a higher risk of disease progression."
In order to convey this information more accurately, I have slightly revised the original text by adding some qualifying language to clarify the scope of this statement. The modified text now reads: "Patients who consistently produce large amounts of urinary protein in many types of kidney disease are also often considered to be at high risk for kidney disease progression."
- The title of second section should be rewritten "A new insight focused on protein particle size and charge for renal injury identification from the perspective of physiological mechanism of GFB" _ in the section were described the data known so far, without any novelty.
Answer: Regarding your inquiry about the lack of novelty in the second part of the content, I fully understand your concern regarding the innovation of the content. However, it is important to emphasize that our team is currently pioneering the field of developing new fluorescence sensors based on urine protein particle size and charge to identify the progression of kidney injury on a global scale. Allow me to briefly explain through the following points:
Methodological novelty: We are introducing a classification method for the first time, which is based on urine protein particle size and charge, and utilizing this method to develop a novel fluorescence sensor. This approach preliminarily enables rapid and accurate identification of kidney injury by amplifying differences in traditional urinary protein indicators, as detailed in our published results in this field (Lines 255-284). This methodological innovation lies at the core of our research and remains an area exclusively explored by our team worldwide.
Novelty in perspective: Our study delves into exploring physiological mechanisms of GFB to investigate correlations between urinary protein size and charge with kidney injury. This unique perspective from basic physiological mechanisms to clinical application offers a fresh understanding of kidney injury while presenting new possibilities for early diagnosis and treatment.
Novelty in practical application: Our preliminary research results demonstrate that our fluorescence sensor based on particle size and charge exhibits characteristics such as speed, convenience, accuracy, providing valuable reference information for doctors in practical clinical applications. Moreover, in our upcoming research publication, we have successfully integrated artificial intelligence algorithms into the detection of clinical chronic kidney disease urine samples which Further increase the clinical application value.
In the second part, we will provide a comprehensive analysis of the mechanical and charge-barrier properties of GFB and their impact on the development of proteinuria. Our objective is not limited to a mere review of existing knowledge, but rather to offer new perspectives for clinical application by leveraging these well-established physiological mechanisms.
- The sentence: Traditionally, protein selectivity has been determined by assessing immunoglobulin G and transferrin clearance [46], - in my opinion it was not traditionally. lt was proposed more than 55 years ago, but not recommended.
Answer: I acknowledge your perspective that the approach in question is not a commonly practiced tradition, and was initially proposed over 55 years ago, but has not gained widespread recommendation. Upon reviewing my statement, I recognize that the use of the word "traditionally" may be somewhat arbitrary. Therefore, I have revised the sentence to read as follows: The determination of protein selectivity has been based on assessing immunoglobulin G and transferrin clearance for an extensive period [46], although this method is rarely employed in clinical settings today.
- The authors underline, that in recent years, there has been rapid development in fluorescence sensing technology, particularly in the field of medical diagnosis. The utilization of fluorescence sensor facilitates the conversion of microscopic biological information into macroscopic visible optical signals-the examples should be provided.
Answer: "In recent years, there has been rapid development in fluorescence sensing technology, particularly in the field of medical diagnosis. The examples of this can be found below (lines 241-254). Since this study focuses on kidney injury, the examples cited are all studies of fluorescence sensors in the field of kidney. They cover different types of fluorescence sensing technologies (such as nanoindicators, fluorescence probes, ratio probes, reporter genes, etc.) and different medical diagnostic applications (such as diabetic nephropathy, drug-induced nephrotoxicity, chronic kidney disease stage, acute kidney injury, etc.), thus demonstrating the universality and diversity of fluorescence sensing technologies.
"The utilization of fluorescence sensor facilitates the conversion of microscopic biological information into macroscopic visible optical signals." All research examples in Section 3.2 of this paper (lines 241-284) are to varying degrees examples of transforming microbiological signals in the body into visual optical signals. This is especially evident in our team's research (lines 255-284) on fluorescence sensors based on urine protein particle size charge. All aim to transform the microbiological signals of urine protein in the body into macroscopic visible optical signals for rapid and sensitive identification of kidney injury processes.
- For me it is a mistake that authors mention traditional Chinese medicine and do not discuss other immunosupressants as calcineurin inhibitors, mycophenolate mofetil or cyclophosphamide in proteinuria (glomerulonephritis) treatment. Rituximab is also not mentioned. Also from the other side for biomedical substances, they only have few literature references.
Answer: Firstly, we fully comprehend your concern for precision in drug selection. The primary objective of this study was to concentrate on clinical agents that can directly target the GFB to reduce proteinuria. Therefore, we intentionally chose classical drugs such as ACEI and ARB, which have been proven to directly act on GFB and reduce proteinuria. The mechanism of these drugs is relatively clear and widely utilized in the clinic, aligning closely with our research goals. Regarding the other immunosuppressants you mentioned, such as calcineurin inhibitors, mycophenolate, cyclophosphamide and rituximab, we acknowledge that these drugs also play a crucial role in the treatment of kidney disease, particularly in autoimmune and tumor-associated nephropathy. However, the mechanism of action of these drugs primarily involves indirectly protecting kidney function by suppressing the immune response or anti-tumor effects rather than directly targeting GFB. For this reason, we have deliberately omitted detailed discussion of these drugs in order to maintain focus and clarity in this discussion.
Regarding the mention of traditional Chinese medicine, we consider it as an acknowledgment of the value of traditional medicine. With thousands of years of accumulated wisdom, traditional Chinese medicine has gained extensive experience in treating kidney disease. Its unique holistic concept and principle of syndrome differentiation provide a valuable supplement to modern medicine. We mention Chinese medicine with the aim of offering readers a more comprehensive perspective and stimulating discussion on new ideas for targeting the GFB to reduce proteinuria.
In conclusion, the selection and discussion of drugs in our paper are based on careful consideration of research purpose and focus.
- Authors anticipate further exploration into this promising new target of the GFB, which will lay a solid foundation for developing novel therapies. In fact there are speculations not supported by any evidence in this manuscript.
Answer: We extend our sincere gratitude to the reviewer for your meticulous evaluation and invaluable suggestions. We fully comprehend and concur with the reviewer's emphasis on the imperative need to further investigate GFB as a therapeutic target. The discussion presented in this paper is not mere speculation, but rather grounded in a profound understanding of GFB, which serves as the fundamental physiological mechanism of kidney function. As an integral component of renal blood filtration and maintenance of internal homeostasis, any alteration in the physiological attributes of GFB is intricately linked to kidney injury.
While this article does not encompass all the experimental data and clinical evidence directly supporting GFB as a new therapeutic target, our perspective is grounded in widely recognized scientific theory and extensive research findings in the current field. There are several authoritative publications, such as Daehn, I.S. and Duffield, J.S., whose article “The glomerular filtration barrier: a structural target for novel kidney therapies” in Nature Reviews Drug Discovery (2021, 20, 770-788) clearly delineates the central role of GFB dysfunction in various kidney diseases and explores the potential for treatment through repairing or modulating GFB function. These studies not only provide substantial evidence to support our perspective but also stimulate our anticipation for future research directions. As we state in the paper (lines 308-309), "most kidney diseases are characterized by GFB dysfunction, making GFB a potential structural target for novel kidney therapies". This viewpoint references Daehn, I.S. and Duffield, J.S.'s article in Nature Reviews Drug Discovery (2021, 20, 770-788). We eagerly anticipate more researchers joining this field in the future to collectively explore the mysteries of GFB and further advance the development of kidney disease treatment.
Round 2
Reviewer 2 Report
Comments and Suggestions for Authors
Unfortunately authors do not answered most of my suggestions.
Currently, kidney injury is classified into AKl and CKD - I do not agree with such comment. For instance - proteinuria, hematuria and accompanying disease syndromes should be also treated as kidney injury.
I do not agree with authors answer. I stand by my opinion.
The title of second section should be rewritten "A new insight focused on protein particle size and charge for renal injury identification from the perspective of physiological mechanism of GFB" _ in the section were described the data known so far, without any novelty.
I do not agree with authors answer. I stand by my opinion.
Novelty in perspective – it is not valid nowadays.
For me it is a mistake that authors mention traditional Chinese medicine and do not discuss other immunosupressants as calcineurin inhibitors, mycophenolate mofetil or cyclophosphamide in proteinuria (glomerulonephritis) treatment. Rituximab is also not mentioned. Also from the other side for biomedical substances, they only have few literature references.
I do not agree with authors answer. I stand by my opinion. For instance calcineurin inhibitors also have non-immunological mechanism of action to decrease proteinuria..
Authors anticipate further exploration into this promising new target of the GFB, which will lay a solid foundation for developing novel therapies. In fact there are speculations not supported by any evidence in this manuscript.
Authors responded to my statement by themselves. “We eagerly anticipate more researchers joining this field in the future to collectively explore the mysteries of GFB and further advance the development of kidney disease treatment.”